

# Rapid surface water volume estimations in beaver ponds

Daniel J. Karran[1], Cherie J. Westbrook[1], Joseph M. Wheaton[2], Carol A. Johnston[3], Angela Bedard-Haughn[4]

[1]Department of Geography and Planning, Centre for Hydrology, University of Saskatchewan, Saskatoon, SK S7N 5C8, Canada
[2]Department of Watershed Sciences, Utah State University, Logan, UT 84322-5210, USA
[3]Department of Natural Resource Management, South Dakota State University, Brookings, SD 57007, USA
[4]Department of Soil Science, University of Saskatchewan, Saskatoon, SK S7N 5A8, Canada

*Correspondence to*: Daniel J. Karran (daniel.karran@usask.ca)

**Abstract.** Beaver ponds are surface water features that are transient through space and time. Such qualities complicate the inclusion of beaver ponds in local and regional water balances, and in hydrological models, as reliable estimates of surface water storage are difficult to acquire without time and labour intensive topographic surveys. A simpler approach to overcome this challenge is needed, given the abundance of the beaver ponds in North America, Eurasia and southern South America. We investigated whether simple morphometric characteristics derived from readily available aerial imagery or quickly measured field attributes of beaver ponds can be used to approximate surface water storage among the range of environmental settings in which beaver ponds are found. Studied were a total of 40 beaver ponds from four different sites in North and South America. The Simplified V-A-h approach, originally developed for prairie potholes, was tested. With only two measurements of pond depth and corresponding surface area, this method estimated surface water storage in beaver ponds within 5% on average. Beaver pond morphometry was characterized by a median basin coefficient of 0.91, and dam length and pond surface area were strongly correlated with beaver pond storage capacity, regardless of geographic setting. These attributes provide a means for coarsely estimating surface water storage capacity in beaver ponds. Overall, this research demonstrates that reliable estimates of surface water storage in beaver ponds only requires simple measurements derived from aerial imagery and/or brief visits to the field. Future research efforts should be directed at incorporating these simple methods into both broader beaver-related tools and catchment scale hydrological models.

## 1 Introduction

The volume of water stored at the surface of wetlands, ponds and lakes is of great concern to those responsible for assessing risks and balancing water supplies to societal demands. Arriving at reliable estimates of such storage is difficult without some knowledge of the feature's morphometry; information that is often time consuming and impractical to acquire, especially when the features are numerous and transient through space and time (Milly et al., 2008). This is particularly true for beaver ponds owing to their cyclic creation and abandonment.





Beaver dams and their associated ponds are ubiquitous in streams and wetlands in the northern hemisphere and southern South America (Westbrook et al., 2013). Beaver dam densities have been reported to exceed 40 dams per kilometer (Macfarlane et al., 2015), making them one of the most frequent obstructions to flowing water (Naiman et al., 1986; Pollock et al., 2003). Although beaver were removed from much of their North American and Eurasian habitat by the fur trade (Naiman et al., 1988),

the past century has seen them steadily re-colonize (Halley et al., 2012; Whitfield et al., 2014). Population models suggest that, given the different climate change scenarios, beaver populations will not only continue to densify, but expand in range (Jarema et al., 2009). Moreover, a further rise in population is likely to be encouraged by recent trends in landscape restoration (Wohl et al., 2015) wherein beaver are being used as a river restoration tool by virtue of the fact that it is a soft engineering approach that is low cost and requires minimal human intervention (Bird et al., 2011; Curran and Cannatelli, 2014; Macfarlane et al.,

2015; Polvi and Wohl, 2013).

A common goal of river restoration is to re-establish the natural flow regime and geomorphic function of the system (Poff et al., 1997; Wohl et al., 2015). Within this context, some beneficial aspects of beaver ponds include an expansion of the riparian zone (Marshall et al., 2013; Pollock et al., 2007), increased aggradation and lateral stability of eroded streams (Burchsted and

Daniels, 2014; Pollock et al., 2014), and an alteration of the timing and peak delivery of water and sediments (Woo and Waddington 1990; Levine and Meyer 2014). Assessment of these benefits would be better contextualized if coupled with the surface water volumes stored in beaver ponds. This would also facilitate a proper risk assessment of using beaver as a river restoration tool, since it is not uncommon for the presence of beaver to be viewed as burdensome or even dangerous from an anthropomorphic perspective. For example, public managers and landowners frequently complain about damage to private

property and infrastructure from beaver-induced flooding (McKinstry and Anderson, 1999; Siemer et al., 2013). Public fears centre around reports of beaver dam failures causing human death and widespread damage from the release of large volumes of water and sediment (Butler and Malanson, 2005; Green and Westbrook, 2009). Such concerns reinforce the need for an easy to use, yet accurate method of estimating the volume of surface water stored in beaver ponds.

As surface water storage is a critical component of any water balance, numerous hydrological investigations have sought to estimate it in other types of wetlands (Trigg et al., 2014; Xu and Singh, 2004). The challenge for hydrological modellers is finding an approach that overcomes the need for often time-intensive topographic surveys; one that is more practical for use in models at varying scales and locations. Previous studies have set about this by defining statistical relationships between surface area and volume for wetlands of specific physiographic regions (Gleason et al., 2007; Hubbard, 1982; Lane and

D'Amico, 2010; Wiens, 2001). Such approaches have been found useful for modelling entire watersheds (Gleason et al., 2007), but limited for estimating storage in individual wetlands because depth and basin morphometry is not considered (Huang et al., 2011; Lane and D'Amico, 2010; Wiens, 2001). Brooks and Hayashi (2002) presented an equation that includes depth and basin morphometry, but in order to use it, basin morphometry must be predefined and no such information yet exists for beaver ponds.





Another approach, the Simplified Volume-Area-Depth (V-A-h) method (Hayashi and van der Kamp, 2000), accounts for depth and calculates basin morphometry for each individual wetland. With little additional effort, it has been shown to provide reliable estimates of surface water storage in the pothole wetlands of the North American prairies for which it was designed (Minke et al., 2010). Prairie potholes are depressional wetlands that have fairly regular shapes, i.e. concave profiles with smooth slopes. Beaver ponds, by contrast, typically encompass a bathymetry that is far more complex because their size and shape is controlled by the dimensions of the dam and the land surface that becomes flooded upon dam establishment (Johnston and Naiman, 1987). Whether statistical or analytical approaches can reliably estimate water storage in beaver ponds has yet to be determined. Our goal was thus to explore tools useful for estimating surface water storage in beaver ponds. We studied beaver ponds across much of their habitat range and: i) evaluated the utility of the Simplified V-A-h method in estimating surface water storage; and, ii) described beaver pond morphometry in relation to water storage capacity.

## 2 Methods

### 2.1 The Simplified V-A-h method

The Simplified V-A-h method is based on a simple power equation (Hayashi and van der Kamp, 2000), where the area of a pond ($A$), at a given height above the pond bottom ($h$), is described as:

$$A = s\left(\frac{h}{h_0}\right)^{2/p}, \tag{1}$$

where $h_0$ is the unit height of the water surface (e.g. 1 m for SI units), $s$ is a scaling coefficient that represents the area of a circle ($m^2$) with a radius that corresponds to $h_0$, and $p$ is a dimensionless morphometry coefficient that represents the shape of the basin profile. The volume of the pond is then determined by integrating all the area profiles below $h$ to give:

$$V(h) = \int_0^h s\left(\frac{h^*}{h_0}\right)^{2/p} dh^* = \left(\frac{s}{1 + 2/p}\right)\left(\frac{h^{1+2/p}}{h_0^{2/p}}\right). \tag{2}$$

Using Eq. (1) and Eq. (2) requires parameterizing the $s$ and $p$ coefficients. The Simplified V-A-h method arrives at these values by rearranging Eq. (1) to give (Minke et al., 2010):

$$s = A_1\left(\frac{h_1}{h_2}\right)^{-2/p}, \tag{3}$$

And

$$p = 2\left(\frac{Log\left(h_1/h_2\right)}{Log\left(A_1/A_2\right)}\right), \tag{4}$$




where $A_1$ and $A_2$ are surface areas of the pond at corresponding depths of $h_1$ and $h_2$, respectively, and $h_1 < h_2$. With only two measurements of area and depth in time, Eq. (3) and Eq. (4) can be used to calculate s and $p$ coefficients that are then reinserted into Eq. (1) and Eq. (2) to define the entire area-depth and volume-depth relationship of the pond.

## 2.2 Beaver pond morphometry

A beaver pond's capacity to store surface water is defined simply by its bathymetry, and can be directly calculated if an accurate topographic survey is available. The problem here relates to how well we can approximate that volume given some simpler measures of the dam dimensions and the physiographic setting in which it exists. That is, can simple measurements, obtained from the field and/or readily available aerial images, be used as an alternative to time and labour intensive topographic surveys? In an effort to describe beaver ponds, a series of morphometric variables were generated from each in addition to the
$p$ coefficient described in Eq. (1).

Key characteristics of any beaver pond are the dimensions of its dam and the corresponding shape of the pond area inundated. The maximum dam height ($h_{max}$) was defined as the difference in elevation (m) between the dam crest elevation and the lowest point in the pond. The length (m) of the dam ($D_{len}$) is measured along its crest. The shape of the beaver pond surface was
described using a dimensionless Shape Index ($S_I$), which is essentially the ratio of the pond perimeter to the circumference of a circle with the same area. It is defined as (Hutchinson, 1957):

$$S_I = \frac{P}{2\sqrt{\pi A_{max}}},\tag{5}$$

where $P$ is the perimeter of the pond (m) and $A_{max}$ is the surface area (m$^2$) at $h_{max}$. Ponds with an $S_I = 1$ have shapes that are perfectly circular, whereas ponds with $S_I > 1$ are increasingly complex. Pond shape is an important metric as much of the interaction between surface water and groundwater happens at the shoreline (Shaw and Prepas, 1990). Previous studies have
used the perimeter-to-area ratio to describe the shape of forest wetlands (Brooks and Hayashi, 2002) and the boundary conditions of beaver ponds (Johnston and Naiman, 1987). We chose $S_I$ for its ease in interpretation and because it enables a relative comparison between the shapes of beaver ponds and other types of wetlands, such as those reported by Minke et al. (2010).

The bathymetric curve (i.e. the area-depth relationship of the pond) is equivalent to the hypsometric curve defined by Strahler (1952) as the ground surface area of a landmass with respect to elevation. In order to compare curves for ponds of different size and relief, it is necessary to express the variables as relative depth ($R_D$) and relative area ($R_A$) as:

$$R_D = \frac{h}{h_{max}},\tag{6}$$

and





$$R_A = \frac{a}{A_{max}}, \qquad (7)$$

where, $h$ is the stage (m) elevation of the pond and $a$ is the corresponding surface area (m$^2$) at any given $h$. For ease of visual interpretation, we express the bathymetric curve as:

$$R_D = (1 - R_A). \qquad (8)$$

Power functions described by Eq. (1) can then be rearranged and expressed independent of the $s$ coefficient as:

$$R_D = (1 - R_A)^{p/2}, \qquad (9)$$

where the $p$ coefficient here is equal to the $p$ coefficient in Eq. (1). This allows for the respective power and bathymetric curves of each pond to be superimposed over one another, thereby eliminating issues of scale and aiding in the analysis of error.

From the relative bathymetric curve, it is possible to compute the Bathymetric Integral ($B_I$), a modified form of the hypsometric integral defined as the measure of landmass volume with respect the entire reference solid created by the maximum dimensions of the pond (Strahler, 1952):

$$B_I = \frac{V_{land}}{h_{max}A_{max}} = \int_0^1 R_A \, dR_D. \qquad (10)$$

Equation 10 produces values between 0 and 1, with 1 representing a reference solid entirely composed of landmass. Using the $B_I$, we introduce a new metric that represents the pond's bathymetric capacity to store water ($B_{WC}$). Since the total volume of the reference solid is 1, the $B_{WC}$, relative to the reference solid, is expressed as:

$$B_{WC} = 1 - B_I = \frac{V_{max}}{h_{max}A_{max}}, \qquad (11)$$

where, $V_{max}$ is the volume of space (m$^3$) that can potentially be filled with water within the reference solid. The $B_I$ and $B_{WC}$ are quantitative measurements of the pond's morphometry and capacity to store water, respectively. The value in using these metrics is that they facilitate the comparison of surface water storage capacity among beaver ponds and other wetland types.

**2.3 V-A-h models for surface water storage estimation in beaver ponds**

Three versions of the power function model described by Eq. (1) were tested in this study. They are referred to as the *Full, Simplified,* and *Optimized* models. The *Simplified* model is the actual test of the Simplified V-A-h method and the other two models were included to aid in the analysis of this approach.

The *Full* model is a power function fitted to the complete dataset of each pond's bathymetry (i.e. empirical fit). We arrive at values for $s$ and $p$ by fitting a simple power function, $y = ax^b$, to the pond's bathymetric curve, and assume $a = s$ and $b = 2/p$ in accordance with Eq. (1). Non-linear least squares regression was used to determine the best-fit; the ability of this model to make accurate area and volume estimates depends on its 'goodness of fit' to the dataset. Analysis of the *Full* model was





included to: i) identify the *p* coefficient that best describes each beaver pond's morphometry; and, ii) assess the overall suitability of power functions to describe beaver pond bathymetry.

The *Simplified* model is a power function using *s* and *p* coefficients created from the same two relative measurements of depth
(i.e. $h_1$ and $h_2$ as a percentage of $h_{max}$) in each pond. Minke et al. (2010) evaluated the Simplified V-A-h method by applying it to two scenarios: a dry one where $h_1$ and $h_2$ are taken at 0.1 m and 25% of $h_{max}$, and a wet one where $h_1$ and $h_2$ are taken at 50% and 75% of $h_{max}$. They found that estimation errors were lowest using the wet scenario; therefore, we chose this scenario to simulate the application of the Simplified V-A-h method as it may be practically used in the field.

The *Optimized* model differs from the *Simplified* model through parameterizing coefficients via the optimum combination of $h_1$ and $h_2$ for each pond. This required calculating *s* and *p* coefficients at every possible combination of $h_1$ and $h_2$ along the bathymetric curve (Note: $h_1$ and $h_2$ are expressed as a percentage of $h_{max}$ from 1–100; therefore, the total number of combinations where $h_1<h_2$ is 5000 for each pond). Each set of *s* and *p* coefficients was then reinserted into Eq. (1) and Eq. (2) to estimate area and volume, respectively, and the set that produced the least combined area and volume error was selected as
the optimum. The *Optimum* model was included in this study to discover how best to use the Simplified V-A-h method with regards to differences in pond morphometry.

Error for all three models was evaluated using root mean square error ($E_{RMS}$), defined as:

$$E_{RMS} = \sqrt{\frac{1}{m}\sum_{i=1}^{m}(D_{ACT} - D_{EST})^2} \, , \qquad (12)$$

where *m* is the number of data points, $D_{ACT}$ is the actual *V-h* relationship or point on the bathymetric curve calculated from the
pond itself, and $D_{EST}$ is the estimated *V-h* relationship or point on the bathymetric curve derived from the *s* and *p* coefficients at a given combination of $h_1$ and $h_2$. Finally, to allow for coherent comparisons of error among the different beaver ponds, the magnitude of error, referred to as $A_{ERR}$ (%) for area and $V_{ERR}$ (%) for volume, was calculated by dividing the $E_{RMS}$ by the actual area and volume of the pond at 80% of $h_{max}$. This particular depth was chosen to avoid inconsistencies in error magnitudes that arise when the evaluation depth is set too close to the minimum and maximum (Minke et al., 2010).

**2.4 Test sites**

Forty beaver ponds were selected for this study and simulated in digital elevation models (DEMs). Our sample design captured the range of structures built by beaver along streams with mineral and organic substrates in both mountainous and lowland terrain. Beaver ponds were thus analyzed from multiple locations where bathymetric data existed, which included: Kananaskis Provincial Park, Alberta, Canada; Escondido, Tierra del Fuego, Argentina; the Logan River watershed, Utah, USA; and,





Voyageurs National Park, Minnesota, USA. Details of the location, terrain, number of ponds, survey methods, and survey resolution for each site are provided in (Table 1).

The beaver ponds in Kananaskis are in the Sibbald research wetland; a rich, flow-through fen formed in an unconstrained basin
of the southern Canadian Rocky Mountains (Westbrook and Bedard-Haughn, 2016). The site is dominated by shrubs (e.g. *Salix* spp. and *Betula* spp*.*), sedges (*Carex* spp.), brown mosses (e.g. *Drepanocladus* spp. and *Scorpidium* spp*.*), and clusters of stunted black spruce (*Picea mariana*), all of which form the peat that reaches depths of up to 6.5 m. Air photo analysis confirms that beaver returned to the basin in the 1950s after their likely extirpation from the region as a result of the fur trade (Morrison et al., 2014). Half of the 10 beaver ponds were surveyed with a Leica total station in 2009 and the other half were
surveyed with a Leica GS15 real time kinetic GPS (rtkGPS) in summer 2015.

Escondido is a southern beech (*Nothofagus pumilio*) dominated forest situated between the well-studied Escondido raised bog and Lago Fagnano on *Isla Grande*, Tierra del Fuego, Argentina. The site is a functioning peatland, dominated by *Sphagnum magellinicum*, in a 150-300 m wide valley that lacks a defined stream channel and instead supports diffuse surface flowpaths
(Westbrook et al. submitted). Beaver are an invasive species in South America, and were purposely introduced to the region in 1946 (Lizarralde et al., 2004). Beaver have since excavated and built low dams composed of peat to form many ponds throughout the length of the valley, hydrologically connecting the Escondido bog to Lake Fagnano. Bathymetry was acquired in three abandoned and drained ponds in the valley during a field visit in February 2013. Surveys of each pond were conducted using a Leica GS15 rtkGPS.

The beaver ponds in the Logan River Watershed are from three different tributaries to the Logan River, namely Spawn Creek, Temple Fork, and Right Hand Fork (Lokteff, 2014; Lokteff et al., 2011, 2013). These sites all occupy partly-confined valley settings with valley fills and a stepped floodplain morphology reflecting centuries of beaver damming. Valley fills are comprised largely of beaver pond deposits and beaver meadows. A few of the ponds are isolated, but most are part of larger
dam complexes consisting of between three and six beaver dams each. Vegetation in the riparian zones and meadows are typically dominated by various assemblages of shrubs (e.g. *Salix* spp.) and herbaceous species (e.g. *Carex* spp. and *Poa* spp.) (Hough-Snee et al., 2013). The beaver ponds were primarily surveyed with a Leica TS15 total station in 2010. Bathymetry was entirely acquired by a Leica TS15 total station and in some localities a Leica 1200 rtkGPS (Lokteff et al., 2011).

Voyageurs National Park lies 35 km east of International Falls, Minnesota, on the U.S.-Canada border. Its Precambrian bedrock is part of the Laurentian Shield, with many outcrops of biotite schist and granite. The soils of beaver ponds are derived from loamy glacial till and clayey glacio-lacustrine sediments that overlie the bedrock, as well as from post-glacial peat deposits (Johnston, 2001). The primary upland vegetation is aspen–birch/boreal conifer forest (*Populus tremuloides, Betula papyrifera, Abies balsamea*). Grasses and sedges (*Calamagrostis canadensis, Carex* spp.) grow in the meadows that form in drained beaver




ponds. Beavers occupied this landscape as of the earliest aerial photos (1927), and the beaver population density peaked at 1.4 colonies/km$^2$ in the 1980s (Johnston and Windels, 2015). Pond DEMs were obtained from light detection and ranging (LiDAR) data flown in May 2011 (MDNR, 2011).

## 2.5 DEM creation and manipulation for variable calculations

Sites selected for this study were former beaver ponds that had drained sufficiently to either reveal pond bottom bathymetry or allow it to be surveyed. Beaver ponds extracted from LiDAR, when available, were fully drained with visible relic dams, whereas some ponds surveyed by total station and rtkGPS often were still full with water up to their crest elevations, but not enough to impede point collection by wading. DEMs that relied on total station and rtkGPS surveys were created with Surfer® v10 (Golden Software, Colorado) using ordinary kriging. The beaver ponds were then isolated from the unneeded areas of the
DEM by extracting all of the points in the raster below and upstream of the dam crest (i.e. $h_{max}$). This was done in ArcGIS v10.2 (ESRI, 2015) as was the calculation of the morphometric variables. The $V$-$h$ relationship and bathymetric curve of each pond was calculated at 5 cm increments using a script written in Python™ that utilizes the 'volume' feature of ArcGIS Toolbox. The $V$-$h$ relationship and bathymetric curve of each pond were the primary inputs for the three models, which were built and run in R Studio (RStudio Team, 2015). Finally, the bathymetric curve for each pond was established using linear interpolation
to create 100 points, i.e. 1–100% of $h_{max}$.

## 3 Results

### 3.1 Beaver pond morphometry

Pond morphometric characteristics are provided in Table 2 and examples of the DEMs from each location are provided in Fig. 1. The 40 ponds well represented the various types of beaver ponds that are created in riverine and wetland habitats (Baker
and Hill, 2003), with max dam heights ($h_{max}$) ranging from 0.25–2 m and dam lengths ($D_{len}$) spanning 3–308 m, with medians of 0.83 m and 40 m, respectively. Pond volumes ($V_{max}$) ranged between 1–9,001 m$^3$ and showed strong power correlations to $D_{len}$, $h_{max}$ and $A_{max}$ (Fig. 2). Among the ponds, there was considerable variability in shape as $S_I$ values ranged from 1.5–5.3 (mean = 2.6). No strong correlations (i.e. -0.10 > R$^2$ < 0.10) were found between $S_I$ and the other morphometric variables used in this study (i.e. $p$, $B_I$, $B_{WC}$, $D_{len}$, $h_{max}$).

The $p$ coefficients for the beaver ponds followed a log normal distribution, and ranged from 0.45–2.58 (median of 0.91) (Fig. 3). Of the 40 beaver ponds analyzed, 70% (28) had $p$ coefficients that were <1, indicating that beaver ponds tend to have convex bathymetries. The majority of beaver ponds tended to be more convex than they are concave, given the shape of the bathymetric curves (Fig. 4) and the range of $B_I$ (0.45–0.85; median of 0.69). In all but one case, $V_{land}$ was greater than 50% of
the total volume of space, indicating that most beaver ponds are shallow, which limits the volume of surface water they can store. This phenomenon is well described by the strong exponential relationship between the $p$ coefficient (R$^2$ = 0.96) and $B_I$



and $B_{WC}$ (Fig. 5). Soil substrate type (Table 1; organic vs. mineral) did not affect the value of the $p$ coefficient, as evidenced by a $t$-test ($P = 0.97$).

## 3.2 Surface water storage estimations

The *Full* model had the least $A_{ERR}$ and the *Optimized* model had the least $V_{ERR}$ (Fig. 6; Table 3). The highest $A_{ERR}$ and $V_{ERR}$
was associated with *Simplified* model estimates, which also produced the greatest variability of error among the different ponds. With regards to study locations, *Full* $V_{ERR}$ ranked as Escondido<Voyageurs<Logan<Kananaskis, whereas *Full* $A_{ERR}$ ranked Logan<Escondido<Kananaskis<Voyageurs. Overall, the beaver ponds in Kananaskis proved most difficult to model (i.e. highest $V_{ERR}$ and $A_{ERR}$ overall); however, mean error for the *Full* model remained below 5% for both area and volume estimates.

Compared to the *Full* model (Fig. 6), the *Simplified* model had higher $V_{ERR}$ in 65% of cases (26 ponds) and higher $A_{ERR}$ in 98% of cases (39 ponds), whereas the *Optimized* model had lower $V_{ERR}$ in 100% of cases but slightly (<1%) higher $A_{ERR}$ in 100% of cases. The optimum $p$ coefficients for volume tended to be slightly different than the optimum $p$ coefficients for area, which are the coefficients derived from the empirical fit of the *Full* model. The *Optimum* model proved useful for revealing the two
points on the bathymetric curve that can be used to obtain the optimum $p$ coefficient for volume estimates. Pond 7 had the largest $A_{ERR}$ and $V_{ERR}$ (Fig. 6), and so was selected for more detailed study (Fig. 7). The optimum points were found at the approximate location of where the empirical fit intersects with the bathymetric curve. Thus, using the optimum points in Eq. (4) computes a $p$ coefficient that is closest to the same coefficient generated by the curve fitted by non-linear least squares regression. The points used by the *Simplified* model for Pond 7 fall on segments of the bathymetric curve that are farther away
in distance from the empirical fit; hence, the $p$ coefficient generated by these points creates a curve that is farther away from the bathymetric curve, which ultimately leads to a less accurate estimate of volume.

In a number of ponds, the empirical fit nearly overlapped the entire bathymetric curve, and in such cases, there were a large number of combinations of $h_1$ and $h_2$ that produced reasonable estimates of volume. For example, Pond 10 had the lowest *Full*
$A_{ERR}$ and $V_{ERR}$ of all the beaver ponds. In this case, there were 1899 combinations of $h_1$ and $h_2$ that produced estimates with total error below 5%, and the distance between the points ranged from 1% to 84% of $h_{max}$. Overall, the error was not sensitive to distance between $h_1$ and $h_2$ as long as the points were on or near the *Full* fitted curve. That said, the average minimum and maximum for $h_1$ (for all of the optimum combinations for each pond) was 18–74%, respectively, and for $h_2$, it was 42–98%, respectively.





## 4 Discussion

The Simplified V-A-h method estimated surface water storage in the beaver ponds with high accuracy. Also, strong statistical relationships were found between surface water storage capacity in beaver ponds and the dimensions of the dam and pond. The beaver ponds studied have a convex shape that permits less water storage than do other open water wetland types. Water storage capacity in beaver ponds varies through time due to beavers' manipulation of their environment. Surface water storage estimates can be made in beaver ponds without the need for topographic surveys if pond morphology is used instead.

### 4.1 V-A-h model performance in beaver ponds

The low *Full* $A_{ERR}$ and $V_{ERR}$ overall indicates that beaver pond morphometry is adequately described by power functions. This is because the bathymetric curve proved resilient to fluctuations in 'elevation' inherent to the impounded land surface. Also, the dams, intricate canals and holes that beaver create in the areas they inhabit (Hood and Larson, 2015) do not warp the shape of the bathymetric curve enough that a power function becomes inappropriate to sufficiently describe it. However, it appears that volume estimations are more resilient to aberrations in the bathymetric curve than are area estimates. The power functions in the *Full* model are fitted to pond bathymetry. When the power curve moves up and down, $A_{ERR}$ will increase, but sometimes the $V_{ERR}$ can decrease because volume is the integration of everything above the bathymetric curve. When the curve moves slightly up or down from the empirical fit, irregularities on the bathymetric curve are captured, which improves volume estimations at the sacrifice of area estimations. This explains why the *Optimum p* coefficients for volume are different than they are for area. It also explains why, in many cases, the *Simplified* model had $V_{ERR}$ that was less than 10%, while $A_{ERR}$ was greater than 25%. Without a complete set of pond bathymetry, it is unlikely that users of the Simplified V-A-h method would be able to discern the optimum points for $h_1$ and $h_2$; however, as long the chosen values for $h_1$ and $h_2$ are selected within the range identified here (i.e. 18–74% of $h_{max}$ for $h_1$ and 42–98% of $h_{max}$ for $h_2$), fairly accurate estimates of surface water storage should be expected. Overall, the *Simplified* model performed reasonably, exceeding 10% $V_{ERR}$ in only three cases. Given that the Simplified V-A-h method appears to work well across the broad range of beaver pond bathymetry reported here, and across a wide range of prairie potholes (e.g. Minke et al., 2010), it should be a robust enough approach to be used other open water wetlands.

### 4.2 Beaver pond morphometry and surface water storage capacity

Our results show that *p* coefficients in beaver ponds are lower overall than those reported in prairie wetlands (Hayashi and van der Kamp, 2000) and those reported in forest pools in New England (Brooks and Hayashi, 2002). Because of the strong exponential relationship between *p* coefficients and $B_{WC}$, we can conclude that beaver ponds typically store less water. For example, the prairie potholes studied by Hayashi and van der Kamp (2000) had a median *p* coefficient of 3.22. Using Fig. 5, this *p* coefficient is equivalent to a $B_{WC}$ of 0.61, which is almost double the median beaver pond $B_{WC}$ equivalent of 0.32. The most likely explanation for this is the ontogeny of beaver ponds compared to other open water wetland types. Beaver ponds





occur via inundation of an existing channel and adjacent riparian area surface, whereas prairie potholes are bowl shaped geomorphic depressions created by the deposition of glacial till (Richardson et al., 1994). These different origins are reflected in the shape of the bathymetric curves, and they also explain the strong statistical relationships between surface water storage capacity and the dimensions of the dam and pond. The stream channel in Fig. 2, for example, is represented on the far right

side of the bathymetric curve. Beaver ponds built on deeper and narrower stream channels tend to have lower $p$ coefficients than ponds built on wider, less constrained channels. This happens because there is a rapid expansion of surface area inundated as the dam exceeds the height and width of the stream channel; a phenomenon that is well described by the 'power' relationships between $D_{len}$, $h_{max}$, $A_{max}$ and $V_{max}$. Pond 12 is a good example of this; the $p$ coefficient was highest (2.58) and a distant outlier compared to the other ponds. The uniqueness of this site is that the beaver built a small dam (0.3 m) with

excavated peat and impounded groundwater seepage rather than damming channel flows. Despite the fact that the dam is relatively small, it has a large $B_{WC}$ (0.55) relative to the other ponds because the dam is entirely dedicated to impounding a mostly flat land surface. In contrast, Pond 6, which was also built in a peatland, has a much lower $B_{WC}$ (0.26) because most of the dam height (2 m) is dedicated to impounding water in an incised stream channel. An advantage of using the $B_{WC}$ metric over pond volumes is that it allows for a comparison of surface water storage capability in a way that is independent of pond

size and shape.

### 4.3 Why surface water storage in beaver ponds varies through time

There are a wide variety of reasons that water storage in beaver ponds vary through time. Some are the result of hydrologic and geomorphic processes, whereas others are the direct result of the ecosystem engineering activity of beaver and/or lack thereof. Among the most common factors include: 1) partial dam breaches by floods (Woo and Waddington, 1990); 2)

aggradation of sediments from upstream sources (Butler and Malanson, 1995); 3) inactive/passive lowering of water surface by dam degradation (Woo and Waddington, 1990); 4) active manipulation of dam height and extent by beaver; and, 5) the excavation of extensive channel networks by beaver (Hood and Larson, 2015). Channel excavation, for example, increases the cumulative area at lower depths of the bathymetric curve, and so increases both the $p$ coefficient and $B_{WC}$. This equates to an increase in surface water storage and we would expect mature ponds (Woo and Waddington, 1990), where beaver have been

active for many years, to have higher $p$ coefficients overall than ponds that were recently built in the same environment. However, the extent by which excavation impacts surface water storage may be partially offset by sediment aggradation, which averages 0.26–2.54 cm/yr in riverine environments (Butler and Malanson 1995). Ponds can thus accumulate large volumes of sediment relative to their surface area (Naiman et al., 1986), which would, over time, increase $B_I$ and decrease the amount of surface water storage in the pond. Studies that explore the influence of beaver pond age on surface water storage are needed.


An interesting consideration is the significance of this variation through time. For any single beaver dam, the results could be dramatically different, but if the population of beaver ponds sampled reflects ponds at various stages of development (Woo & Waddington 1990), the temporal variation is presumably captured in the spatial variation by sampling over a larger area. For





individual beaver ponds, it would be worth testing whether or not the simple methods presented herein still work reasonably well if dam lengths and dam heights are measured or specified to reflect actual stage instead of just the crest stage. It may be that the variation in water surface storage could be reasonably estimated by considering different crest elevations. Such an analysis may allow exploration of the role of active beaver maintenance on water storage and we would postulate that active maintenance increases such storage. We have observed, in many systems, that a very high quantity of beaver dams are actually 'maintained' by very few beaver that move around quite regularly. Such mobile beaver should spend less time maintaining water surface elevations than those that stay locally and just maintain one complex and lodge. The water storage benefit of beaver ponds in areas that are very under-seeded with beaver and below population capacity (even if at dam capacity), is that the benefit is minimized where beaver populations are not close to carrying capacity.

## 4.4 Tools for surface water storage estimation in beaver ponds

There are a variety of ways our results can be used to estimate surface water storage in beaver ponds under different data availability scenarios. In situations where only aerial or remotely sensed imagery is available (i.e. world-wide), dam length and pond area can be approximated and used in the generalized power regression relationships presented in Fig. 2. This is a quick and easy way to incorporate beaver pond surface water storage capacity into land-use planning decisions and watershed-scale hydrological models. However, this approach is not suitable for detailed study in individual beaver ponds as it does not account for pond morphometry (Huang et al., 2011; Wiens, 2001). Including dam height should improve estimates. Measuring dam height in the field is quick and straight forward, but it can also be reasonably approximated with remotely sensed imagery alone using spectral-depth correlation methods (e.g. Passalacqua et al., 2015). If dam heights are available, we recommend using our median $p$ coefficient (0.91) for beaver ponds in the equation presented by Brooks and Hayashi (2002):

$$V_{max} = \frac{A_{max} \times h_{max}}{1 + {}^{2}\!/p}. \tag{13}$$

This equation is a modified form of Eq. (2) used to estimate surface water storage capacity. It is easily incorporated into spatially distributed hydrological models. Fang et al. (2010) had success in using this approach, albeit for prairie potholes, in their Cold Regions Hydrological Model.

With a moderate amount of data, the Simplified V-A-h method offers an alternative that produces surface water storage estimates with minimal error. The advantage of this method over the others is that it is robust, it is customized to each pond's basin morphometry, and it calculates a coefficient of scale (i.e. $s$ coefficient) for use in estimating surface water storage across the range of pond stages, unlike the generalized power regression models and Eq. (13), which are limited to estimates of $V_{max}$. Combined with a few field visits and something as simple as automated water level observations, the Simplified V-A-h method can be a powerful tool. But, it also has practical application in relatively data rich environments. For example, many LiDAR datasets are collected when beaver ponds are not fully drained. If the beaver pond is not entirely full, the measurements for $A_2$ and $h_2$ can be measured within the vertical distance between the crest of the dam and the surface of the water, thus allowing



an appropriate $p$ coefficient to be derived. Furthermore, the Simplified V-A-h method is increasingly practical with the advent of new technologies. For example, Structure from Motion software facilitates the creation of high resolution DEMs from ordinary photographs (Javernick et al., 2014). Theoretically, with both of these tools, one field visit to collect a few pictures and depths measurements should be all that is needed to make reliable estimates of wetland surface water storage.

## 4.5 Implications of study results

The results of our study provide some simple tools that enable surface water storage in beaver ponds to be estimated without the need for topographic surveys. This allows environmental managers to better assess the risks and benefits associated with beaver ponds that appear on landscapes, and allows the easy inclusion of the surface water storage component of beaver ponds into hydrological models at various scales. This study also demonstrates that beaver pond morphometry is different than other types of wetlands, which requires consideration. For example, based on this analysis we might expect beaver ponds to reach their capacity faster during rainfall events, while impounding larger surface areas than depressional wetlands. Although we show that some beaver ponds store less surface water than other wetland types, their relevance to local and regional water balances should not be underestimated. Beaver population recovery, post fur trade, has led to the creation of between 9,494–42,236 km$^2$ of new beaver ponds globally (Whitfield et al. 2014). Using Whitfield et al.'s (2014) estimates and our median $p$ coefficient (0.91) and median dam height (0.83 m) in Eq. (13), we crudely estimate that between 2.5 and 11 km$^3$ of water are stored in beaver ponds.

## 5 Conclusions

The primary goal of this study was to test the utility of readily applicable tools for estimating surface water storage in beaver ponds. We examined whether the Simplified V-A-h method was appropriate for this purpose and described beaver pond morphology to explore its relationship to surface water storage capacity. A number of valuable insights were revealed. The Simplified V-A-h method proved to be a simple and effective tool as it was able to estimate beaver pond surface water storage with an average volume error of 5%. The median basin coefficient for beaver ponds was 0.91, suggesting that they tend to have a convex basin morphometry, and that they typically store less water than other wetlands studied in the same way. Pond capacity was strongly correlated to the dimensions of the dam and surface area of the pond, further cementing the idea that beaver ponds exhibit characteristic traits in pond morphometry that make reliable estimates of surface water storage possible without the need for topographic surveys. Future research efforts should be directed at applying these simple methods more remotely, and incorporating them into both broader beaver-related planning tools and catchment-scale hydrological models.

*Acknowledgements* We thank Carlie Elliott, Kirsten Allen, Kenny DeMeurichy, Ryan Lokteff, and Konrad Hafen for field assistance. We also thank the staff and students of the Centre for Hydrology, particularly Sebastian Krogh, Chris Marsh, and Phillip Harder for their assistance with computer programming. We would also like to thank the University of Calgary




Biogeoscience institute for accommodation and access to their facilities. The project was funded through a Natural Sciences and Engineering Research Council (NSERC) of Canada Doctoral Graduate scholarship, an NSERC Discovery Grant (RGPIN 32837-20), the Global Institute of Water Security at the University of Saskatchewan, and a National Science Foundation Grant (DEB-1349240). The Utah field data collection was funded by the United States Forest Service (Cooperative Agreement 09-

CS-1300001885391 awarded to Utah State University: Award 100030) and facilitated by USFS partner Brett Roper. Research in Argentina was conducted under SDSyA Resolution #414/2013 and facilitated by Christopher Anderson and Alejandro Valenzuela.

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





**TABLES**

**Table 1** Site locations, characteristics and details of topographic pond surveys.

| Site | Latitude and Longitude (°,') | n | Soil Substrate Type | Terrain | Survey method | DEM resolution (m) |
|---|---|---|---|---|---|---|
| Kananaskis Provincial Park, AB, Canada | 51° 3.553' N, 114° 52.009' W | 10 | Organic | Mountainous | rtkGPS | 1 |
| Escondido, Tierra del Fuego, Argentina | 54° 36.908' S, 67° 44.540' W | 3 | Organic | Mountainous | rtkGPS | 1 |
| Logan River Watershed, UT, USA | 41° 50.327' N, 111° 33.668' W | 14 | Mineral | Mountainous | Total Station | 0.1 |
| | 41° 49.568' N, 111° 34.516' W | 2 | Mineral | Mountainous | Total Station | 0.1 |
| | 41° 48.868' N, 111°35.553' W | 5 | Mineral | Mountainous | Total Station | 0.1 |
| Voyageurs National Park, MN, USA | 48˚ 32.773' N 93˚ 4.328' W | 1 | Organic | Lowland | LIDAR | 1 |
| | 48˚ 27.975' N 92˚ 53.864' W | 3 | Mineral | Lowland | LIDAR | 1 |
| | 48˚ 30.405' N 92˚ 40.331' W | 1 | Mineral | Lowland | LIDAR | 1 |
| | 48˚ 31.262' N 92˚ 52.794' W | 1 | Mineral | Lowland | LIDAR | 1 |





**Table 2** Pond morphometric characteristics, including the shape index ($S_I$), bathymetric integral ($B_I$), bathymetric water capacity ($B_{WC}$), length of the dam ($D_{len}$), and maximum depth ($h_{max}$), area ($A_{max}$), and volume ($V_{max}$) of the ponds.

| Location | Pond # | $S_I$ | $B_I$ | $B_{WC}$ | $p$ | $s$ (m²) | $D_{len}$ (m) | $h_{max}$ (m) | $A_{max}$ (m²) | $V_{max}$ (m³) |
|---|---|---|---|---|---|---|---|---|---|---|
| Kananaskis | 1 | 2.05 | 0.75 | 0.25 | 0.69 | 889 | 164 | 1.50 | 2974 | 1135 |
| | 2 | 2.37 | 0.77 | 0.23 | 0.61 | 356 | 152 | 1.75 | 2006 | 867 |
| | 3 | 2.57 | 0.69 | 0.31 | 0.97 | 959 | 127 | 0.85 | 686 | 186 |
| | 4 | 1.79 | 0.77 | 0.23 | 0.61 | 123 | 27 | 1.50 | 446 | 163 |
| | 5 | 3.71 | 0.77 | 0.23 | 0.62 | 705 | 226 | 1.95 | 5496 | 2503 |
| | 6 | 3.47 | 0.74 | 0.26 | 0.67 | 1845 | 199 | 2.00 | 16357 | 9001 |
| | 7 | 1.76 | 0.76 | 0.24 | 0.56 | 1334 | 308 | 1.85 | 12912 | 5734 |
| | 8 | 2.55 | 0.75 | 0.25 | 0.63 | 701 | 159 | 1.80 | 3787 | 1757 |
| | 9 | 1.71 | 0.68 | 0.32 | 0.92 | 290 | 39 | 1.25 | 448 | 184 |
| | 10 | 1.51 | 0.66 | 0.34 | 1.05 | 247 | 30 | 1.10 | 297 | 113 |
| Escondido | 11 | 2.32 | 0.59 | 0.41 | 1.16 | 5352 | 162 | 0.55 | 1528 | 325 |
| | 12 | 1.72 | 0.45 | 0.55 | 2.58 | 2181 | 59 | 0.30 | 748 | 130 |
| | 13 | 1.99 | 0.54 | 0.46 | 1.61 | 3223 | 124 | 0.55 | 1342 | 344 |
| Logan | 14 | 2.19 | 0.66 | 0.34 | 1.06 | 438 | 7 | 0.30 | 54 | 6 |
| | 15 | 2.03 | 0.72 | 0.29 | 0.83 | 464 | 3 | 0.25 | 15 | 1 |
| | 16 | 1.89 | 0.56 | 0.44 | 1.51 | 87 | 4 | 0.60 | 41 | 11 |
| | 17 | 2.63 | 0.75 | 0.25 | 0.67 | 112 | 17 | 0.75 | 52 | 10 |
| | 18 | 2.14 | 0.70 | 0.30 | 0.91 | 91 | 19 | 0.80 | 63 | 15 |
| | 19 | 2.17 | 0.67 | 0.33 | 0.97 | 138 | 10 | 0.65 | 53 | 11 |
| | 20 | 1.95 | 0.67 | 0.33 | 0.94 | 352 | 16 | 0.45 | 50 | 8 |
| | 21 | 2.47 | 0.64 | 0.36 | 1.11 | 179 | 11 | 0.50 | 45 | 8 |
| | 22 | 2.70 | 0.67 | 0.33 | 0.98 | 96 | 7 | 0.45 | 17 | 2 |
| | 23 | 1.90 | 0.64 | 0.36 | 1.20 | 56 | 10 | 0.55 | 23 | 5 |
| | 24 | 1.97 | 0.69 | 0.31 | 0.80 | 430 | 27 | 0.60 | 82 | 15 |
| | 25 | 2.37 | 0.59 | 0.41 | 1.36 | 154 | 6 | 0.30 | 22 | 3 |
| | 26 | 2.83 | 0.75 | 0.25 | 0.68 | 124 | 21 | 0.90 | 90 | 19 |
| | 27 | 2.79 | 0.73 | 0.27 | 0.75 | 114 | 5 | 0.60 | 36 | 6 |
| | 28 | 1.73 | 0.67 | 0.33 | 0.96 | 278 | 13 | 1.00 | 265 | 87 |
| | 29 | 4.32 | 0.81 | 0.19 | 0.45 | 975 | 87 | 1.00 | 980 | 189 |
| | 30 | 3.43 | 0.71 | 0.29 | 0.79 | 620 | 21 | 0.85 | 374 | 94 |
| | 31 | 5.31 | 0.69 | 0.31 | 0.90 | 551 | 43 | 0.85 | 432 | 115 |
| | 32 | 2.61 | 0.69 | 0.31 | 0.83 | 1647 | 46 | 0.50 | 210 | 32 |
| | 33 | 2.59 | 0.66 | 0.34 | 0.99 | 409 | 51 | 1.65 | 1123 | 621 |
| | 34 | 2.40 | 0.58 | 0.42 | 1.41 | 470 | 12 | 0.45 | 130 | 26 |
| Voyageurs | 35 | 4.65 | 0.71 | 0.29 | 0.83 | 3683 | 144 | 1.10 | 4725 | 1517 |
| | 36 | 3.82 | 0.70 | 0.31 | 0.94 | 4539 | 161 | 1.10 | 5928 | 1999 |
| | 37 | 3.54 | 0.72 | 0.28 | 0.78 | 36105 | 57 | 0.40 | 2297 | 264 |
| | 38 | 2.52 | 0.66 | 0.34 | 0.88 | 11836 | 58 | 1.10 | 12985 | 4740 |
| | 39 | 2.72 | 0.61 | 0.39 | 1.11 | 18033 | 97 | 0.90 | 12482 | 4350 |
| | 40 | 2.78 | 0.63 | 0.37 | 1.18 | 4867 | 41 | 0.55 | 1504 | 316 |





**Table 3** V-A-h model performance comparisons based on the mean (± standard deviation) volume ($V_{ERR}$) and area ($A_{ERR}$) error magnitude

| Site | n | Full | | Simplified | | Optimized | |
|---|---|---|---|---|---|---|---|
| | | $V_{ERR}$(%) | $A_{ERR}$ (%) | $V_{ERR}$(%) | $A_{ERR}$ (%) | $V_{ERR}$(%) | $A_{ERR}$ (%) |
| Kananaskis | 10 | 4.3 ± 3.1 | 3.8 ± 2.1 | 7.2 ± 6.0 | 14.6 ± 12.3 | 2.3 ± 1.6 | 4.2 ± 2.5 |
| Escondido | 3 | 3.1 ± 1.4 | 3.8 ± 0.7 | 4.3 ± 2.5 | 6.7 ± 3.8 | 1.6 ± 0.7 | 4.0 ± 0.9 |
| Logan | 21 | 4.0 ± 2.6 | 3.6 ± 1.7 | 4.6 ± 3.5 | 9.9 ± 8.5 | 2.0 ± 1.2 | 3.9 ± 1.9 |
| Voyageurs | 6 | 3.8 ± 1.8 | 4.1 ± 1.6 | 3.8 ± 1.8 | 4.1 ± 1.6 | 1.9 ± 0.9 | 4.4 ± 1.7 |
| **All Ponds** | **40** | **4.0 ± 2.5** | **3.8 ± 1.7** | **5.2 ± 4.1** | **11.0 ± 9.4** | **2.1 ± 1.2** | **4.0 ± 1.9** |





**Fig. 1.** Four examples of detrended beaver pond DEMs used for this study, one from each study area. ($S_I$ = shape index, $B_I$ = Bathymetric integral, $B_{WC}$ = Bathymetric water capacity, $p$ = Basin coefficient, $s$ = Scaling coefficient, $D_{len}$ = Dam length, $h_{max}$ = Maximum height of the dam, $A_{max}$ = Maximum surface area of the pond, and $V_{max}$ = Maximum volume of the pond)



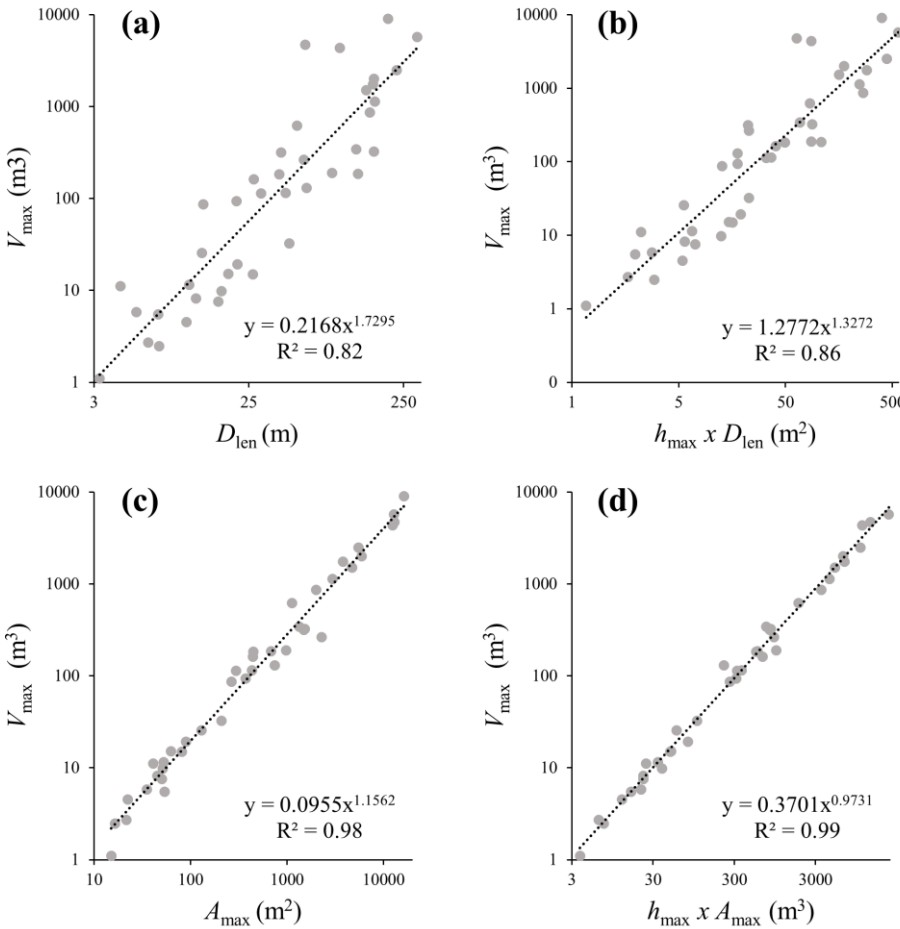

**Fig. 2.** Power regression relationships between the maximum volume of the beaver ponds ($V_{max}$) and: (a) the length of the beaver dams ($D_{len}$);
10    (b) the product of the maximum depth of the ponds ($h_{max}$) and the length of the beaver dams; (c) the maximum surface area ($A_{max}$) of the ponds; and, (d) the product of the maximum surface area and maximum depth of the pond.





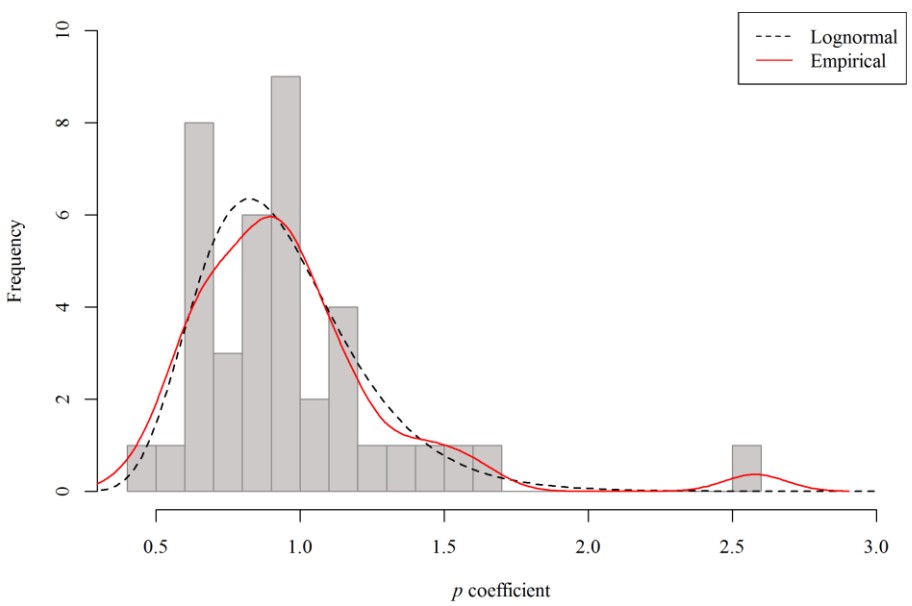

**Fig. 3.** Distribution of *p* coefficients for all beaver ponds sampled (n=40).

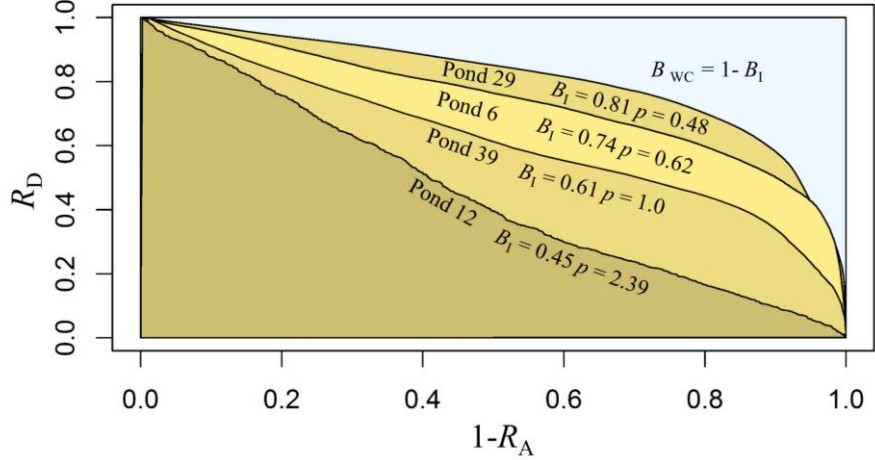

**Fig. 4.** Bathymetric curves for ponds shown in Figure 1.



Hydrology and
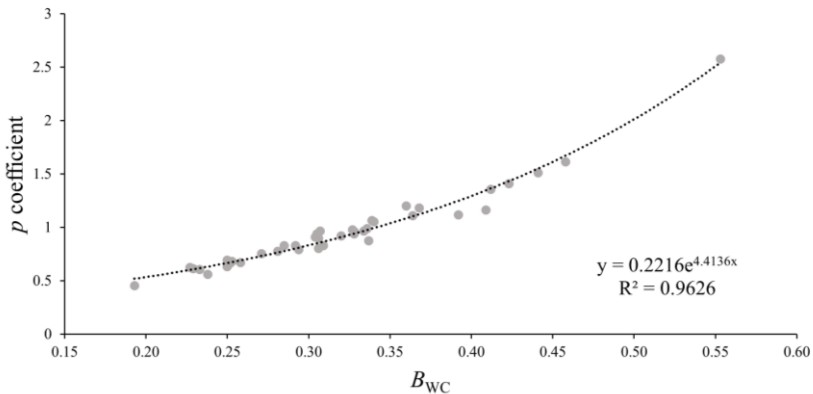

**Fig. 5.** Relationship between the *p* coefficient and *B*WC.





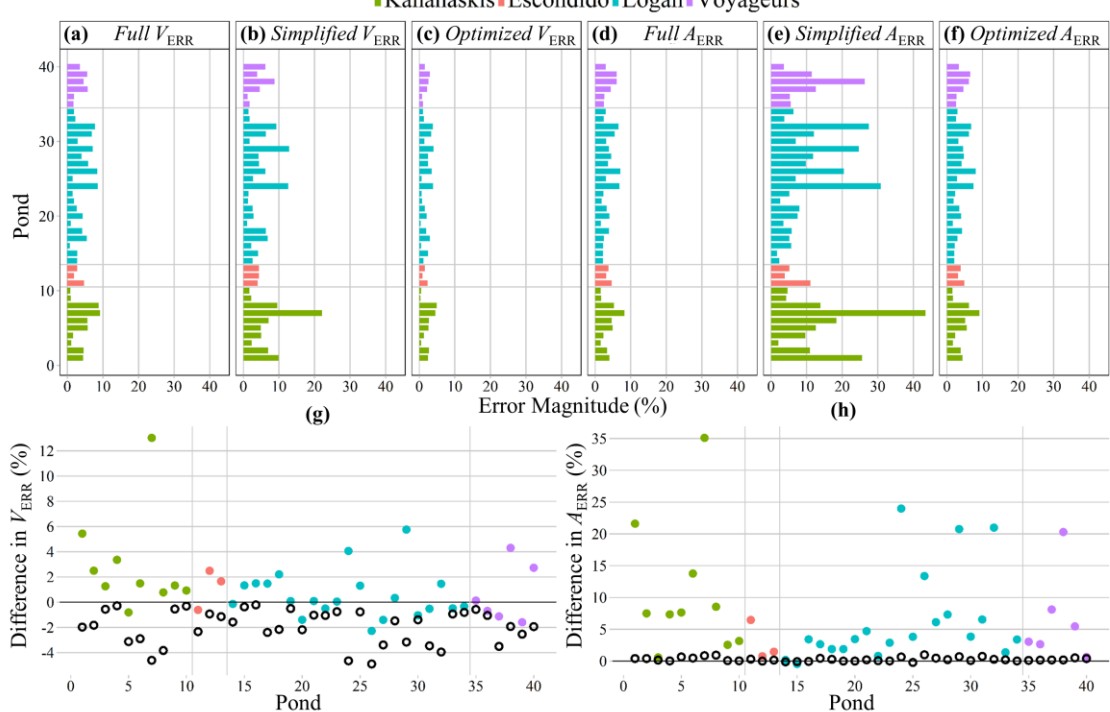

**Fig. 6.** Volume ($V_{ERR}$) and area error ($A_{ERR}$) from each beaver pond using the three different approaches (a-f). Plots on the bottom show the difference in volume (g) and area (h) error of the *Simplified* (solid circles) and *Optimized* (open circles) models relative to the *Full* model (the *Full* model is represented by the solid black line at zero on the y-axis). Bars and solid circles are colour coded by location as per the legend at the top of the figure.



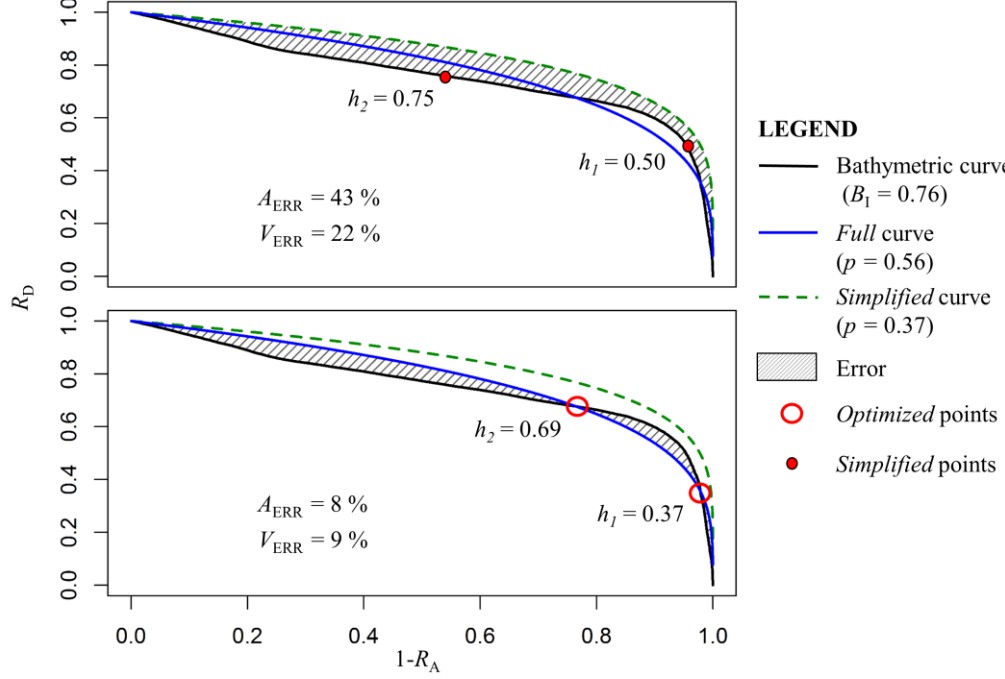

**Fig. 7.** Comparison of the hypsometric curve for Pond 7 with the *Full* and *Simplified* curve. The top shows the error associated with the *Simplified* curve that was calculated using Simplified $h_1$ and $h_2$ and the bottom shows the error associated with the *Full* curve and the optimum location for $h_1$ and $h_2$.

