# Peer review of "Rapid surface water volume estimations in beaver ponds"

_Hydrology and Earth System Sciences, 2016_

## Referee Comment (RC1) · S. Markstrom (Referee) · 21 Sep 2016

Thank you for the opportunity to review the draft manuscript "Rapid surface water volume estimations in beaver ponds" by Daniel J. Karran, Cherie J. Westbrook, Joseph M. Wheaton, Carol A. Johnston, and Angela Bedard-Haughn. I found this article to be a very interesting description of a method for determination of storage in beaver ponds using a minimum of input information. The authors make a compelling argument for this method with examples. I found this draft to be very clean and well written. Please consider the following comments.

1. eqn 8: Just to be clear about where equation 8 came from, add: "For ease of visual interpretation, equations 6 and 7 can be combined as:" 2. eqn 9: Is this correct? If eqn 8 is substituted into eqn 9, you get $RD = RD^p/2$ 3. p 5, ln 5 – I don't understand the

statement "thereby eliminating issues of scale and aiding in the analysis of error." 4. eqn. 10 – define Vland. Also, add more description to the paragraph starting at page 5, line 6. It is hard to understand BI just by reading the text (e.g. what is the "reference solid" that you refer to). I had to refer to Strahler, 1952 to understand this paragraph. 5. table 1 – what is the column "n" in the table? I assume number of ponds at the site.
* * *

---

## Referee Comment (RC2) · Anonymous Referee #2 · 21 Nov 2016

Recommendation: Major Revisions

General Comments:

This study explores the capabilities of different geometric methods to estimate surface water storage in beaver ponds; to do so, the authors use topographic datasets from multiple beaver ponds that range in hydrogeologic setting. The paper is generally well written (but see technical corrections) and presents results in informative, polished figures. The paper's main contribution is its quantitative comparison of different methods, which require different input datasets (from simply dam length to coupled measures of water depth and inundated area), to predict beaver surface water storage (see Section 4.4). Considering the number of beaver ponds and their contribution to watershed storage and release, such assessed tools are useful for watershed assessments, planning, and modeling over large spatial scales. However, I find the paper too long and with unnecessary text for a focused evaluation of methodologies for storage volume estimation. With that said, there are also opportunities (often alluded to in the text) to expand the work, where the focus is broader: beaver pond morphologies, their drivers, and their implications. As such, I see two options to reframe the paper that should be considered:

1) Technical Note: Streamline the paper's text and focus to compare methods for predicting volumes. This will also require clarifying some of the methods and their linkages (see specific comments below). Text to consider removing/shortening includes:

Page 1, Line 25 through Page 2, Line 23. Rather, the introduction could succinctly state: beaver ponds are ubiquitous and important to watershed water storage, highlighting the need for methods to quickly estimate storage use; methods have been developed for other wetland features, and here we apply these for beaver ponds.

Section 2.5. No need to describe the sites in detail (e.g., vegetation); instead, simply present needed information (hydrogeologic setting, DEM datasets) in the table.

Text distributed throughout results (e.g., Page 8, Lines 26-31) that describes the variation in beaver pond morphology. This text should be retained for Option 2 (below), but removed for a technical note solely focusing on a methodology.

Similarly, Sections 4.2 and 4.3 could be removed for a technical note; instead, the discussion should simply revisit the methods to discuss tradeoffs between accuracy and data needs among the methods evaluated (i.e., section 4.4).

2) Research Article: For this option, the manuscript could be expanded, where it focuses on the variation, drivers, and implications of a suite of morphological metrics (in addition to storage volume) for beaver ponds. At times, the manuscript points to some of these topics (e.g., the importance of SI for groundwater exchange, beaver ponds store less water than potholes b/c of ontogeny of ponds, time variation of pond morphometry, etc), but these points seem somewhat tangential for the current manuscript focus. However, the paper could make a meaningful comparison across ponds and regions by deemphasizing the volume storage methodology and including: 1) a full comparison of the different metrics (SI, storage, dam lengths, maximum depth, etc) across systems, 2) analyses of their drivers (e.g., predictive relationships with stream order, watershed slope, etc), and 3) focused intro and discussion text regarding implications (cumulative storage, perimeter to area ratios for water exchange and habitat, sediment storage, etc). Again, there is some mention of such topics (e.g., Section 4.5), but a quantitative evaluation of the drivers and importance of beaver morphology means a full treatise on this subject, where the volume storage estimation is one method applied. For this option, authors could consider either just including the 40 ponds used here (in which case, the actual bathymetric curves could be used), or they could use the 40 ponds to verify the Simple V-A-h method, and then use a larger set of ponds with available required datasets to derive volume, SI, and other metrics.

In short, I contend that the manuscript is lacking clear and organized scope. The two options suggested above will help frame the work in a clear way, be it as a technical note or an evaluation of beaver pond morphologies; I believe either option will provide a valuable contribution. Given this suggested shift in scope, specific comments depend on option chosen. As such, I have limited the number of comments below, and include only those that should be addressed regardless of option, or that I point to an Option-specific revision.

Specific Comments:

Page 1, Line 14: Be specific when discussing surface water storage as a function of stage vs. storage capacity.

Page 2, Line 6: Why are beaver populations expected to increase with climate change?

Page 2, Lines 12 – 23: Too much focus here on restoration, even for Option 2, and especially for a technical note. Instead, informing restoration is just one importance of

good volume estimation.

Page 2, Line 31: Define basin morphometry.

Page 3, Line 2: Qualify "with little additional effort".

Methods: For option 1, a conceptual 2-panel figure (cross section and plan view) would really help to define terms used in the equations.

The methods are hard to follow; some reorganization and explicit text to link methods would help; how this is done will depend on the manuscript's new scope. For Option 1, this would mean revising Section 2.2 to explicitly distinguish the variables that were used for simple predictions of volume (dam length, SI) versus the relationships that were used to evaluate model predictions (i.e., Dact). It could also be clearer what Dact and Dest refer to; the "actual V-h relationship or point on the bathymetric curve" makes that confusing without more clarification. It might help to switch 2.2 and 2.3. For Option 2, methods would reflect the various different metrics used to describe pond bathymetry and how these were compared across sites.

Page 5, Eqns 8 and 9: Where is the exponent in Eqn 8 that then appears as p/2?

Section 2.4: Again this information could be streamlined and probably just included in a table.

Section 4.3: Points raised here are not addressed by the results. For Option 1, remove text altogether, other than just pointing to the importance of a simple method to estimate storage considering this time variability. For Option 2, consider retaining text, but only if some results can point to this time variability.

Page 12, Lines 29-31: Good point and method application.

Section 4.5. Example of inferences that could be expanded in Option 2 but minimized in Option 1.

Technical Corrections:

Page 1 Line 27: Complete sentences should be follow semicolons. This needs to be addressed throughout in a number of places (e.g., Page 2, Line 27)

Page 2, Line 8: "by virtue of the fact that it.." Awkward.

Page 2, Line 32: . . .basin morphometry are not considered.

Page 9, Line 4: Need a comma after Aerr.

---

## Author Response (AR1)

**POINT BY POINT RESPONSE TO REVIEWERS**

Reviewer comments are in blue, our proposed changes are in grey, and the actual changes we made to the document are in red.

**Response to reviewer #1:**

5   Thank you for the opportunity to review the draft manuscript "Rapid surface water volume estimations in beaver ponds" by Daniel J. Karran, Cherie J. Westbrook, Joseph M. Wheaton, Carol A. Johnston, and Angela Bedard-Haughn. I found this article to be a very interesting description of a method for determination of storage in beaver ponds using a minimum of input information. The authors make a compelling argument for this method with examples. I
10  found this draft to be very clean and well written.

Please consider the following comments.1. eqn 8: Just to be clear about where equation 8 came from, add: "For ease of visual interpretation, equations 6 and 7 can be combined as:" 2. eqn 9: Is this correct? If eqn 8 is substituted into eqn 9, you get RD = RDˆp/2 3. p 5, ln 5 – I don't understand the statement "thereby eliminating issues of scale and aiding in the analysis
15  of error."

Equation 9 is used to fit a power function to the actual points on the bathymetric curve, which makes Equation 8 redundant, and its inclusion confusing. Thus, we will remove Equation 8 from the revised manuscript. We will also more plainly state that the intentions of Equations 6-9 are to scale the bathymetric curves to one another to facilitate comparison among them.

20  Equation 8 was removed and the following sentence proceeds what used to be Eq. (9) but is now Eq. (7) in the revised document:

*"Power functions described by Eq. (1) can then be fit to a bathymetric curve with the following equation:"*

We clarified what we intended to convey with regards to scale and error with the following
25  paragraph after Eq. (7)"

*"where the p coefficient here is equal to the p coefficient in Eq. (1). This allows for a visual aid in the analysis of error by superimposing estimated curves produced via either Eq.(1) or Eq. (4) to the pond's actual bathymetric curve. It also eliminates issues of scale between different ponds so that bathymetric curves can be visually compared to one another."*

eqn. 10 – define Vland. Also, add more description to the paragraph starting at page 5, line 6. It is hard to understand BI just by reading the text (e.g. what is the "reference solid" that you refer to). I had to refer to Strahler, 1952 to understand this paragraph.

Vland will be defined in the revised manuscript (Eqn 10). We will also improve our description of BI, probably using a simple example, in the revised manuscript to improve understandability.

We added a perceptual diagram (Figure 1) showing the relationship between all the morphometric variables.

table 1 – what is the column "n" in the table? I assume number of ponds at the site.

Yes, "n" does indeed represent pond numbers. We will add this to the table caption.

We added that "n" represents the number of ponds in the tables captions where it was used.

**Response to reviewer #2:**

This study explores the capabilities of different geometric methods to estimate surface water storage in beaver ponds; to do so, the authors use topographic datasets from multiple beaver ponds that range in hydrogeologic setting. The paper is generally well written (but see technical corrections) and presents results in informative, polished figures. The paper's main contribution is its quantitative comparison of different methods, which require different input datasets (from simply dam length to coupled measures of water depth and inundated area), to predict beaver surface water storage (see Section 4.4). Considering the number of beaver ponds and their contribution to watershed storage and release, such assessed tools are useful for watershed assessments, planning, and modeling over large spatial scales. However, I find the paper too long and with unnecessary text for a focused evaluation of methodologies for storage volume estimation. With that said, there are also opportunities (often alluded to in the text) to expand the work, where the focus is broader: beaver pond morphologies, their drivers, and their implications. As such, I see two options to reframe the paper that should be considered:

Technical Note: Streamline the paper's text and focus to compare methods for predicting volumes. This will also require clarifying some of the methods and their linkages (see specific comments below). Text to consider removing/shortening includes: Page 1, Line 25 through Page 2, Line 23. Rather, the introduction could succinctly state: beaver ponds are

ubiquitous and important to watershed water storage, highlighting the need for methods to quickly estimate storage use; methods have been developed for other wetland features, and here we apply these for beaver ponds. Section 2.5. No need to describe the sites in detail (e.g., vegetation); instead, simply present needed information (hydrogeologic setting, DEM datasets) in the table. Text distributed throughout results (e.g., Page 8, Lines 26-31) that describes the variation in beaver pond morphology. This text should be retained for Option 2 (below), but removed for a technical note solely focusing on a methodology. Similarly, Sections 4.2 and 4.3 could be removed for a technical note; instead, the discussion should simply revisit the methods to discuss tradeoffs between accuracy and data needs among the methods evaluated (i.e., section 4.4).

Research Article: For this option, the manuscript could be expanded, where it focuses on the variation, drivers, and implications of a suite of morphological metrics (in addition to storage volume) for beaver ponds. At times, the manuscript points to some of these topics (e.g., the importance of SI for groundwater exchange, beaver ponds store less water than potholes b/c of ontogeny of ponds, time variation of pond mor-phometry, etc), but these points seem somewhat tangential for the current manuscript focus. However, the paper could make a meaningful comparison across ponds and regions by deemphasizing the volume storage methodology and including: 1) a full comparison of the different metrics (SI, storage, dam lengths, maximum depth, etc) across systems, 2) analyses of their drivers (e.g., predictive relationships with stream order, watershed slope, etc), and 3) focused intro and discussion text regarding implications (cumulative storage, perimeter to area ratios for water exchange and habitat, sediment storage, etc). Again, there is some mention of such topics (e.g., Section 4.5), but a quantitative evaluation of the drivers and importance of beaver morphology means a full treatise on this subject, where the volume storage estimation is one method applied. For this option, authors could consider either just including the 40 ponds used here (in which case, the actual bathymetric curves could be used), or they could use the 40 ponds to verify the Simple V-A-h method, and then use a larger set of ponds with available required datasets to derive volume, SI, and other metrics. In short, I contend that the manuscript is lacking clear and organized scope. The two options suggested above will help frame the work in a clear way, be it as a technical note or an evaluation of beaver pond morphologies; I believe either option will provide a valuable contribution. Given this suggested shift in scope, specific comments depend on option chosen. As such, I have limited the number of comments below, and include only those that should be addressed regardless of option, or that I point to an Option specific revision.

Our goal was to discover tools useful for estimating surface water storage in beaver ponds at large and small spatial scales – ones that are easy to apply in relatively data-sparse

environments, and ones that hold potential for incorporation into hydrological models in future research initiatives. While we agree with reviewer 2 that a full treatise on beaver pond morphology is needed, enhancement of water storage on the landscape owing to biota – in our case beaver damming activities - is the focus of our paper. Indeed, a treatise on beaver pond morphology may not yet be possible. For our research, we contacted all the leading beaver impact researchers across the world and learned pond morphometry is rarely documented as part of their research initiates. Thus, the 40 ponds we studied represent nearly the whole of the global population of beaver ponds with detailed morphometric measurements.

That said, we agree with reviewer 2's suggestion to streamline the paper's text and focus. But, we do not think the streamlining will reduce the text and content enough to align the manuscript with requirements for a technical note. Reduction of the manuscript to a technical note would require solely evaluating the Simplified V-A-h method. Such focus would be of more limited interest and use to readers of HESS as it would eliminate our evaluation and discussion of tools useful for estimating surface water volumes at larger spatial scales. The discussion of tools for estimating surface water storage volumes at larger scales originates through our characterization of pond morphometry.

To shorten the paper, we will, as reviewer 2 suggests, reduce the length of the Introduction. We will re-focus the Introduction to succinctly make the following points: beaver ponds are ubiquitous and important to watershed water storage, highlight the need for methods to quickly estimate storage use; identify methods that have been developed for other types of wetlands, and state how we here apply these to beaver ponds. We also foresee shortening section 2.5 by removing non-critical site detail, such as the description of site vegetation. We plan to retain section 4.2 as it is critical to discussion of our results. Section 4.3 will be removed in its entirety.

We refocused and shortened the introduction by removing the information about using beaver for river restoration and the expansion of beaver habitat due to climate change. We also altered our objectives in accordance with changes to the methods as follows:

*"We studied beaver ponds across much of their habitat range and: i) evaluated the utility of the Simplified V-A-h method in estimating surface water storage; ii) evaluated correlations between surface water storage and beaver pond morphometry; and, iii) described beaver pond morphometry in relation to surface water storage capacity."*

We removed all the non-critical site detail from section 2.4 and removed section 4.3 entirely.

Page 1, Line 14: Be specific when discussing surface water storage as a function of stage vs. storage capacity.

We will make the suggested changes

We define what we mean by surface water storage in the first sentence of the introduction as follows:

*"The volume of water stored at the surface of wetlands, ponds and lakes (as a function of stage) is of great concern to those responsible for assessing risks and balancing water supplies to societal demands."*

Page 2, Line 6: Why are beaver populations expected to increase with climate change?

We will detail that climate change is expected to produce a modest expansion of the northern range limit of beaver by 2055 (Jarema et al., 2009).

We removed this sentence from the manuscript to shorten the introduction

Page 2, Lines 12 – 23: Too much focus here on restoration, even for Option 2, and especially for a technical note. Instead, informing restoration is just one importance of good volume estimation.

We agree that informing restoration is just one importance of good volume estimation and will change the text to say so instead of going into detail on the application of the method to restoration.

We removed this section of the introduction to address the reviewer's comment and shorten.

Page 2, Line 31: Define basin morphometry.

We will follow the example of Brooks and Hayashi (2002) and define basin morphometry parenthetically as "(surface, volume, depth)."

We added this exactly as proposed

Page 3, Line 2: Qualify "with little additional effort".

We will qualify this statement to make it less subjective

We removed "with little additional effort" and changed the sentence to read:

*"Requiring only two measurements of depth and surface area, it has been shown to provide reliable estimates of surface water storage in the pothole wetlands of the North American prairies for which it was designed (Minke et al., 2010)."*

Methods: For option 1, a conceptual 2-panel figure (cross section and plan view) would really help to define terms used in the equations.

We will include a figure like this in the revised paper to help define terms used in the equations.

We added this figure to the manuscript. It is now Figure 1

The methods are hard to follow; some reorganization and explicit text to link methods would help; how this is done will depend on the manuscript's new scope. For Option 1, this would mean revising Section 2.2 to explicitly distinguish the variables that were used for simple predictions of volume (dam length, SI) versus the relationships that were used to evaluate model predictions (i.e., Dact). It could also be clearer what Dact and Dest refer to; the "actual V-h relationship or point on the bathymetric curve" makes that confusing without more clarification. It might help to switch 2.2 and 2.3. For Option 2, methods would reflect the various different metrics used to describe pond bathymetry and how these were compared across sites.

The methods will be modified in accordance to the suggestions provided for Option 1 and addition of a figure, as suggested in the last comment.

We divided the methods into two subsections- "2.2.1 Metrics for surface water volume estimations", and "2.2.2 Morphometric analysis." We streamlined the text in each of these sections to be more concise about what we are trying to achieve with each morphometric variable. Further, we added a perceptual diagram (Figure 1), providing a visual aid for understanding the relationship between variables.

Page 5, Eqns 8 and 9: Where is the exponent in Eqn 8 that then appears as p/2?

As identified by Reviewer #1, there is an error in equation 8, which will be removed from the revised paper and replaced by equation 9.

We removed Equation 8 from the paper and revised that section as noted above.

Section 2.4: Again this information could be streamlined and probably just included in a table.

We will make the suggested changes

We removed all the non-critical site detail. Section 2.4 is now a short paragraph and a table.

Section 4.3: Points raised here are not addressed by the results. For Option 1, remove text altogether, other than just pointing to the importance of a simple method to estimate storage considering this time variability. For Option 2, consider retaining text, but only if some results can point to this time variability.

We will remove section 4.3 entirely as suggested by the reviewer.

We removed this section entirely

Page 12, Lines 29-31: Good point and method application.

Thank you.

Section 4.5. Example of inferences that could be expanded in Option 2 but minimized in Option 1.

Section 4.5 will be fleshed out with examples based on the changes we indicated above.

All the inferences in this section are in accordance with our discussion as revised.

Page 1 Line 27: Complete sentences should be follow semicolons. This needs to be addressed throughout in a number of places (e.g., Page 2, Line 27)

Noted errors were corrected throughout

Page 2, Line 8: "by virtue of the fact that it.." Awkward.

This was removed from the manuscript

Page 9, Line 4: Need a comma after Aerr.

These three grammatical changes will be made.

Noted error was corrected

5  Page 2, Line 32: : : :basin morphometry are not considered.

We will change 'is' following 'basin morphometry' to 'are' to make grammatical sense.

We changed exactly as proposed

**PLEASE FIND THE MARKED UP MANUSCRIPT BELOW**

[revised manuscript text omitted]